# Environmental DNA reveals patterns of biological invasion in an inland sea

Joe Duprey[1]*, Ramón Gallego[1,2], Terrie Klinger[1], Ryan P. Kelly[1]

**1** School of Marine and Environmental Affairs, University of Washington, Seattle, WA, United States of America, **2** Universidad Autónoma de Madrid—Unidad de Genética, Madrid, Spain

* jduprey@uw.edu

**Data Availability Statement:** The minimal underlying dataset and complete analysis code are available at the following public Github repository: https://github.com/jdduprey/patterns_of_invasion. The version of code and data that appears in this publication has been cataloged on zenodo: https://

## Abstract

Non-native species have the potential to cause ecological and economic harm to coastal and estuarine ecosystems. Understanding which habitat types are most vulnerable to biological invasions, where invasions originate, and the vectors by which they arrive can help direct limited resources to prevent or mitigate ecological and socio-economic harm. Information about the occurrence of non-native species can help guide interventions at all stages of invasion, from first introduction, to naturalization and invasion. However, monitoring at relevant scales requires considerable investment of time, resources, and taxonomic expertise. Environmental DNA (eDNA) metabarcoding methods sample coastal ecosystems at broad spatial and temporal scales to augment established monitoring methods. We use COI mtDNA eDNA sampling to survey a diverse assemblage of species across distinct habitats in the Salish Sea in Washington State, USA, and classify each as non-native, native, or indeterminate in origin. The non-native species detected include both well-documented invaders and species not previously reported within the Salish Sea. We find a non-native assemblage dominated by shellfish and algae with native ranges in the temperate western Pacific, and find more-retentive estuarine habitats to be invaded at far higher levels than better-flushed rocky shores. Furthermore, we find an increase in invasion level with higher water temperatures in spring and summer across habitat types. This analysis contributes to a growing understanding of the biotic and abiotic factors that influence invasion level, and underscores the utility of eDNA surveys to monitor biological invasions and to better understand the factors that drive these invasions.

## Introduction

Invasive marine species impose increasingly high economic and ecological costs [1, 2]. They threaten fisheries, aquaculture, and marine recreation [3–5], disrupt food-webs, alter habitat structure, and displace native species [6–8]. Climate change, coastal eutrophication, aquaculture, and maritime trade all facilitate the spread and establishment of non-native species [9–13].

Prevention and early detection of biological invasions leads to the most successful economic and ecological outcomes [14–16]. Towards this goal, the sampling of environmental DNA (eDNA) has shown promise when applied to the detection of non-native species [17]. Targeted

zenodo.org/record/8436270 with the
accompanying DOI: 10.5281/zenodo.8436270.

**Funding:** The authors received no specific funding
for this work.

**Competing interests:** The authors have declared
that no competing interests exist.

quantitative Polymerase Chain Reaction (qPCR) methods have successfully revealed the presence of individual species of interest, including the terrestrial toad species *Bufo japonicus formosus* in Hokkaido, Japan and invasive crayfish in Baden-Württemberg, Germany [18, 19]. Data collected using qPCR has motivated management and legal action on invasive carp in the Great Lakes region of the United States [20]. Sampling and analysis of eDNA data can be used independently or to complement established monitoring methods such as visual surveys. For example, data from eDNA sampling was paired with conventional trap data to better quantify the invasion front of European green crab (*Carcinus maenas*) in the Salish Sea, Washington [21].

At the level of ecological communities, metabarcoding reflects a subset of species present. In the Celtic Sea and the Arctic Ocean, this technique has been used to detect multiple non-native species in water samples collected across a large spatial scale [22, 23]. Metabarcoding has further been employed to monitor the shifting ranges of harmful marine microalgae [24].

In invasive-species research, metabarcoding methods offer advantages of scalability and replicability, at comparatively low cost [25]. By using common "universal" primers such as the 313 COI fragment we use here [26], metabarcoding can sample species across the tree of life that are present at a particular location and point in time. Each metabarcoding assay will depict a sample of the underlying biodiversity [27], and although (as with any sampling method) the species detected are subject to various biases, these biases are driven by primer-template chemistry, and so are unlikely to systematically discriminate between non-native and native species. As such, amplicon sequencing is a powerful way to investigate a host of questions about ecological invasion, including those that require simultaneous data on native and non-native assemblage composition and species richness.

A foundational question in invasion ecology concerns the biotic and abiotic factors that make habitats vulnerable to invasion [28, 29]. Significant progress has been made towards answering this question via visual sampling. Surveys in Northern Europe and San Francisco Bay have shown that relative non-native richness (= non-native species richness / total species richness) tends to be higher in estuarine habitats of meso- and polyhaline salinties (~ 5–25 ppt) than in either fresh and oligohaline habitats (< 5 ppt), or more marine polyhaline and euhaline habitats (> 25 ppt) [30–33]. A global literature review of invertebrate invasion data [31] showed that, with few exceptions, temperate estuaries are more invaded than adjacent open coasts. Proposed explanations for this phenomenon include: (1) most ports occupy mesohaline and polyhaline areas of estuaries; (2) species adapted to mid-range salinities are more likely to survive the variable salinities and temperatures incurred during transport in ballast or as hull biofouling; (3) mid-salinity regions face propagule pressure from both salt-tolerant freshwater species and freshwater-tolerant estuarine species; and (4) mesohaline and polyhaline waters have lower native species richness than either fresh or euhaline waters [33–35].

Ecosystem-scale metabarcoding surveys provide a new lens through which to examine invasion ecology. In particular, this data source can contribute substantially to the debate over biotic resistance—the degree to which native species richness and abundance determines habitat invasibility. Ecologists have hypothesized that biotic resistance works via multiple mechanisms including competition, disease, and predation [36]. Resistance can result from a single key native species, or from the sum of resource utilization dynamics among the entire native assemblage [29]. Experimental, mesocosm, and observational studies have reached conflicting conclusions about the net effect of native species richness on habitat invasibility [37]. For example, the non-native zooplankton species *Daphnia lumholtzi* was more likely to be present in zooplankton mesocosms with higher native richness [38]. Conversely, in seagrass mesocosms, the biomass of non-native invertebrates was shown to have an inverse relationship with native invertebrate richness [39]. Observational field surveys have similarly reached mixed

conclusions [40]. The net effect of native biotic resistance remains unclear, as does its relative importance compared to abiotic factors, the characteristics of potential invaders, and propagule pressure. In human-impacted estuarine ecosystems, aquaculture is an important source of propagule pressure [11, 41]. Past approaches to quantifying the relative effects of habitat invasibility—biotic resistance and abiotic factors—as opposed to propagule pressure have included probabilistic modeling [42, 43] and field manipulations [44].

While previous marine and aquatic metabarcoding surveys have cataloged invasion fronts and the distributions of non-native species, none has systematically investigated invasion levels across a range of coastal habitats. Here, we describe variation in invasion levels in the southern Salish Sea (Washington State, USA) determined using eDNA sequences from discrete water samples collected from eight nearshore sites.

## Methods

### Water sampling

To characterize eDNA assemblages we used data originally collected for the analysis reported in Gallego et al. (2020). Water samples were collected in the intertidal zone of state and county parks along shores of Hood Canal and San Juan Island, Washington, USA (**Fig 1**). In Washington state no permits are necessary for small-scale collection of sea water. Abiotic conditions at these sites fall roughly along a gradient of salinity, temperature, and wave energy, with lower salinity, higher temperature, and lower wave energy at the Hood Canal sites compared with the San Juan Island sites [45]. Substrate type differs across these sites as well: the three southernmost sites along the Hood Canal consist of mudflat habitat, while the two northern sites on the Hood Canal are cobble beaches, and the San Juan Island sites are primarily rocky bench [45].

Three one-liter replicate bottle samples were collected at each of the eight sites monthly between March 2017 and August 2018. Samples were filtered through 0.45 μm cellulose and these filters were preserved in Longmire's buffer prior to DNA extraction [46]. Water temperature, salinity, and dissolved oxygen were measured during each sampling event with a multiprobe (Hannah Instruments, USA) and a salinity refractometer.

### Sequencing and bioinformatics

The DNA was purified with a phenol-chloroform-isoamyl alcohol extraction following procedures in Renshaw et al. (2015) [46]. Extracted DNA served as a template for polymerase chain reaction (PCR), amplifying 313 base pairs of the COI gene region [26]. PCR was conducted using protocols from Kelly et al. (2018) [47]. Three subsamples of the template DNA served as technical process replicates. These technical replicates were amplified separately and sequenced separately to assess the variability of the PCR process itself. Secondary indexing tags were introduced with a two-step PCR protocol in order to avoid index amplification bias [48]. We sequenced multiplexed sampling events using MiSeq v2-500 and v3-600 sequencing kits via manufacturer specifications. Each sequencing run included three samples of terrestrial species DNA not present in the study region–red kangaroo *Macropus rufus* or ostrich *Struthio camelus*–as a positive control to avoid misassignment of sequences to samples, and to control for "index-hopping" [45] "tag-jumping" [49].

Sequencing quality control, denoising into Amplicon Sequence Variants (ASVs) and taxonomical assignment were conducted using custom Bash and R scripts. A Github repository containing these scripts, and access to the corresponding FASTA sequence data is linked in the supplementary material (**S1 File**).

First, we executed Bash scripts to run the open-source programs Cutadapt and DADA2 for primer-trimming and removal of PCR artifacts [50, 51]. DADA2 was used to estimate the ASV

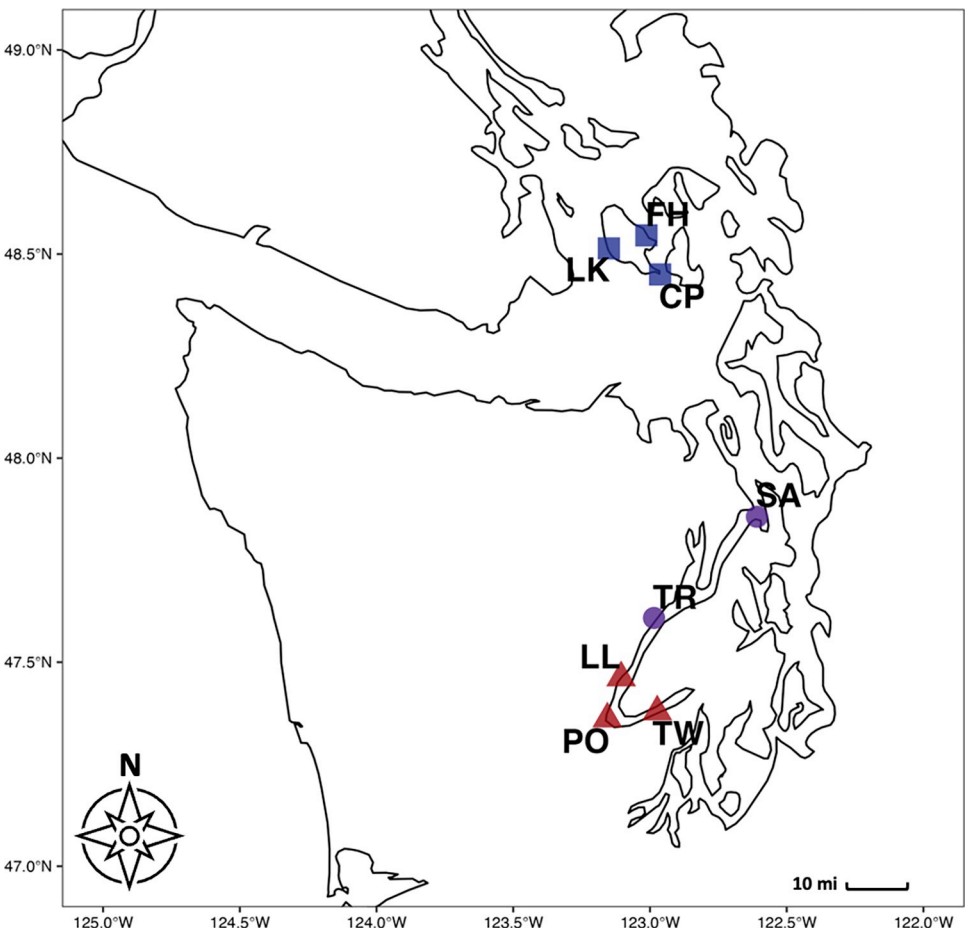

**Fig 1. The southern portion of the Salish Sea, an estuarine fjord of the Northwest Pacific located in Washington State, USA.** Sampling sites are coded by habitat type: red = more-retentive estuarine mudflat, purple = more-retentive estuarine cobble beach, blue = rocky bench.

composition of each sample. Second, we conducted further quality control of the results by discarding replicates with low coverage containing PCR anomalies and spurious ASVs. Full details and rationale can be found in Gallego et al. (2020).

We used two bioinformatic tools to assign taxonomy to ASVs. *Insect* v1.1 was run to assign taxa via informatic sequence classification trees [52]. Additional ASVs were assigned to taxa using a custom COI database with *anacapa* and *Bowtie2* [53, 54]. For the purpose of quantifying the species richness of the eDNA sampled, we retained only those ASVs identified to the species level. We performed a secondary BLASTn search on all ASVs assigned to the species level. Any ASV with a percent sequence identity (pident) < 95% with its best-match reference sequence was considered unclassified. A species was considered present in a sample if an ASV assigned to that species appeared in at least 1 out of 3 replicates for a given site and date. See Gallego et al (2020) for full bioinformatic details.

## Identification of native, cryptogenic, and non-native species

The retained ASVs with species-level assignments were referenced against published literature and peer-reviewed online databases to classify each species as either native to the Salish Sea, non-native to the Salish Sea, or cryptogenic. If World Register of Marine Species (WORMS),

Algaebase, Biodiversity of the Central Coast, or the National Estuarine and Marine Exotic Species Information System (NEMESIS) listed a species as having a distribution containing the Salish Sea; then the species was classified as native [55–58]. If no peer-reviewed source contained the species distribution, or if there were fewer than three published detections of the species, then the species was classified as cryptogenic.

Species were classified as non-native if multiple peer-reviewed sources listed the species as non-native to the Salish Sea or non-native to other temperate estuaries (n = 17). Or, if there was no such record, the species was classified as non-native if all Salish Sea native species within the same genus were present in the NCBI COI database, and the observed ASV both differed from those in the database and matched the non-native unambiguously (n = 2). Where peer-reviewed literature contained contradictory information regarding native distribution, species were classified as non-native if at least three distinct sources supported non-native status in the Salish Sea.

In order to exclude cases of ASVs from poorly cataloged native taxa being incorrectly assigned to non-native sister taxa; ASVs with percent sequence identity < 98% assigned to non-native taxa were included as non-native only if multiple peer-reviewed sources listed the species as non-native to the Salish Sea (n = 2). See supplementary materials for complete BLAST results and classification flowchart (**S1 Fig in S1 File**).

### Testing for invasion-associated variables

To determine which abiotic and biotic factors best predicted non-native species richness, we modeled non-native species richness as a Poisson variable, with the expected value being a function of salinity, temperature, and native species richness; we used R and the rstanarm package [59, 60] to compare model fits using subsets of these candidate predictor variables and chose the best-fit model using leave-one-out cross-validation, which maximizes the out-of-sample predictive value among candidate models [61]. This pool of predictive variables was selected because the invasion ecology literature suggests non-native species richness varies across ranges of salinity, temperature, and native species richness, and because we were able to measure these factors in the field. Consequently, we tested models with combinations of these three factors.

To quantify the variation of non-native species richness and level of invasion across sampling sites and sampling months, we used the vegan package in R to perform permutational multivariate analyses of variance (PERMANOVA) using Euclidean pairwise distances, with site and month as nested grouping factors [62].

## Results

In total, sequencing yielded 50.8 million reads across 86 unique sampling events. Denoising to ASVs and removal of replicates with low coverage retained 45.0 million of these reads made up of 4,848 unique ASVs. Of these ASVs, 1,364 could be annotated to a taxonomic level of family, genus, or species, representing 22.8 million reads. 324 ASVs were assigned to the species level and had a percent identity with the matching reference sequence of > 95%. These 324 ASVs represented 11.0 million retained reads (**Table 1**).

### Composition

After the quality-control process we were able to assign ASVs to 272 species across the 8 sampling sites. Of these 272 species, we identified 251 as native or cryptogenic, and 21 as non-native (**Table 2**). Seven phyla were represented among these non-native species: Rhodophyta, Ochrophyta, Mollusca, Arthropoda, Chordata, Cnidaria, and Annelida. Non-native species

**Table 1. The number of reads in millions, and the number of ASVs retained after each step in the bioinformatic process and subsequent quality control steps.**

| step | reads (millions) | ASVs |
|---|---|---|
| Sequencing, removal of PCR artifacts | 50.8 | NA |
| Denoising to ASV, removal of replicates with low coverage | 45.0 | 4,848 |
| ASVs assigned to family, genus, or species level with *Insect*, *anacapa*, *bowtie2* | 22.8 | 1,364 |
| ASVs assigned to species with Blastn percent identity > 95% | 11.0 | 324 |

included 9 algal species, 5 bivalve species, 3 copepod species, 1 ascidian species, 1 hydrozoan species, 1 polychaete species, and 1 amphipod species.

Two non-native bivalve species—the Manilla clam (*Ruditapes philippinarum*) and the Pacific oyster (*Crassostrea gigas*)—are commonly cultivated in commercial aquaculture operations in Hood Canal. These aquaculture species were detected in 25 of all 86 and 13 of all 86 samples respectively. We note that many species present in the environment are not represented in the data; any given metabarcoding locus amplifies a cross-section of species present, rather than a complete set. Accordingly, species detected here are a subset of those present, with detection bias driven largely by template-primer binding affinity. Hence, nondetection of a species cannot be used to infer that species' absence from the environment without further information about the amplification efficiency of the species with the primer set in question.

## Relative and absolute species richness

Absolute native species richness varied significantly between sites, but not between months (PERMANOVA: site $R^2 = 0.47$, $p < 0.001$; month $R^2 = 0.08$, $p = 0.23$). Absolute non-native species richness varied significantly between both sites and months. (PERMANOVA: site $R^2 = 0.54$, $p < 0.001$; month $R^2 = 0.14$, $p = 0.002$) The majority of non-native species (18 of 21) were detected only at the more-retentive estuarine sites in Hood Canal (Twanoh, Potlatch, Lilliwaup, Triton Cove, Salisbury). By contrast, only 3 of 21 species were detected at the more marine sites (Friday Harbor, Cattle Point, Lime Kiln) on San Juan Island (**Fig 2**). Of these, the hydrozoan *Bougainvillia mucus* was the only species detected exclusively at the San Juan Island sites. Two non-native species were shared between Hood Canal and San Juan Island sites: the clam *Nuttallia olivacea* and the harmful alga *Pseudochattonella farcimen*, which was the most frequently detected non-native species.

Relative non-native richness was calculated as *non-native species richness / total species richness*. Relative non-native richness varied significantly between both sites and months (PERMANOVA: site $R^2 = 0.56$, $p < 0.001$; month $R^2 = 0.17$, $p < 0.001$). At the San Juan Island sites, relative non-native richness was negligible across the months sampled. Relative non-native richness was greater at the Hood Canal sites, peaking at 0.25 (8 / 32) at the Lilliwaup site in August 2017. We found substantial variation in relative non-native richness at each of the Hood Canal sites across sampling events (**Fig 3**). Relative non-native richness was higher during the warmer, more productive spring and summer months (May-September), and lower during the cooler months (October-March) **Figs 4 and 5**.

The mean richness of non-native species was low at the San Juan Island Sites (Friday Harbor = 0, Cattle Point = 1, Lime Kiln = 1) compared to the Hood Canal sites (Twanoh = 4, Potlatch = 4, Lilliwaup = 4, Triton Cove = 5, Salisbury = 2). The mean richness of native species was greater at the San Juan Island sites (Friday Harbor = 48, Cattle Point = 58, Lime Kiln = 62) than it was at the Hood Canal sites (Twanoh = 32, Potlatch = 33, Lilliwaup = 30, Triton Cove = 34, Salisbury = 38) (**Fig 6**).

**Table 2. Non-native species recorded at least once across all 86 sampling events.**

| species | Detections | BLASTn pident | e value | Matching nucleotides |
|---|---|---|---|---|
| Pseudochattonella farcimen | 59 | 100 | $7.14e^{-161}$ | 313 / 313 |
| Gracilaria vermiculophylla | 34 | 100 | $7.14e^{-161}$ | 313 / 313 |
| Ruditapes philippinarum | 25 | 100 | $7.14e^{-161}$ | 313 / 313 |
| Caulacathus okamurae | 23 | 100 | $7.14e^{-161}$ | 313 / 313 |
| Nuttallia olivacea | 17 | 100 | $7.14e^{-161}$ | 313 / 313 |
| Crassostrea gigas | 13 | 100 | $6.96e^{-156}$ | 304 / 304 |
| Callithamnion corymbosum | 13 | 100 | $7.14e^{-161}$ | 313 / 313 |
| Neoporphyra haitanensis | 12 | 99.68 | $5.56e^{-157}$ | 312 / 313 |
| Monocorophium acherusicum | 9 | 100 | $7.14e^{-161}$ | 313 / 313 |
| Gelidiophycus freshwateri | 9 | 100 | $7.14e^{-161}$ | 313 / 313 |
| Botrylloides violaceus | 6 | 100 | $7.14e^{-161}$ | 313 / 313 |
| Bougainvillia muscus | 3 | 100 | $7.14e^{-161}$ | 313 / 313 |
| Lomentaria hakodatensis | 3 | 100 | $7.14e^{-161}$ | 313 / 313 |
| Mya arenaria | 3 | 100 | $1.55e^{-162}$ | 316 / 316 |
| Gelidium vagum | 2 | 100 | $7.14e^{-161}$ | 313 / 313 |
| Musculista senhousia | 2 | 96.16 | $7.35e^{-141}$ | 301 / 313 |
| Nitokra spinipes | 2 | 98.91 | $3.44e^{-134}$ | 309 / 313 |
| Hediste diadroma | 1 | 99.68 | $4.27e^{-158}$ | 310 / 311 |
| Melanothamnus harveyi | 1 | 98.72 | $3.35e^{-154}$ | 310 / 313 |
| Mytilicola orientalis | 1 | 97.77 | $1.70e^{-92}$ | 306 / 313 |
| Stenhelia pubescens | 1 | 100 | $7.14e^{-161}$ | 313 / 313 |

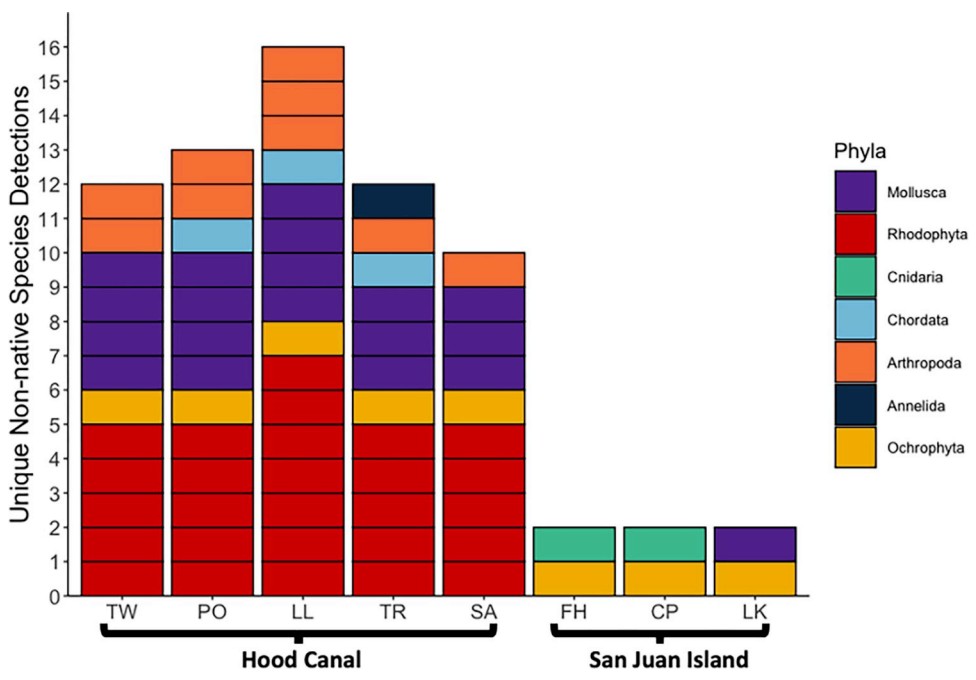

**Fig 2. Number and phyla of unique detections of non-native species at the eight sampling sites.** For each site, if a species was detected at least once during the 18-month sampling period it is displayed as a unique detection.

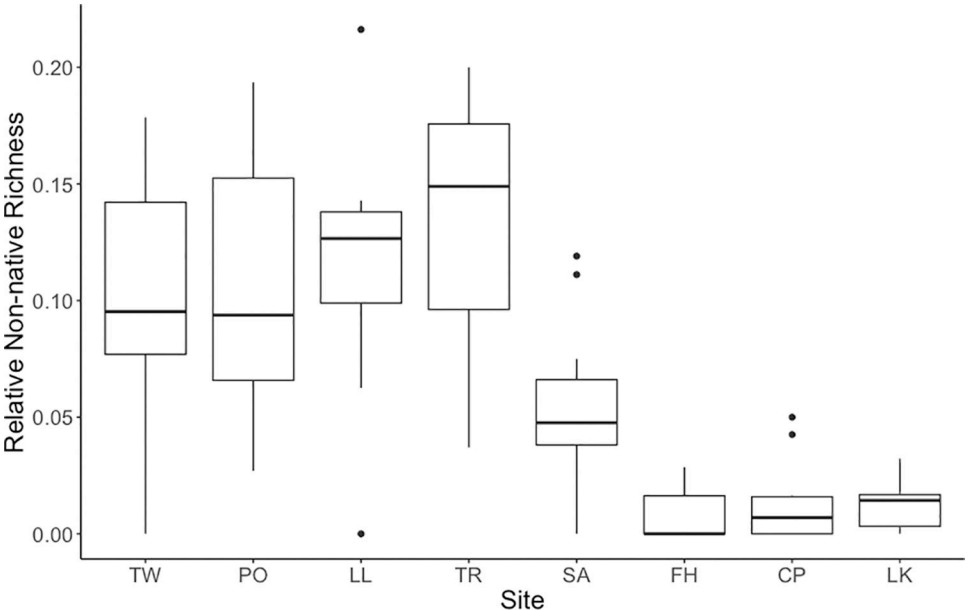

**Fig 3. Boxplots of relative non-native richness (non-native species richness / total species richness) across the eight sampling sites.** Sites are ordered from the southernmost more-retentive estuarine sites on the left, to the northernmost less-retentive rocky bench sites on the right.

## Invasion level across biotic and abiotic conditions

Our sampling captured a broad range of water temperatures (min = 7.17 C, max = 22.60 C) (Fig 5), salinity (min = 10.00 ppt, max = 30.31 ppt) and native species richness (min = 13,

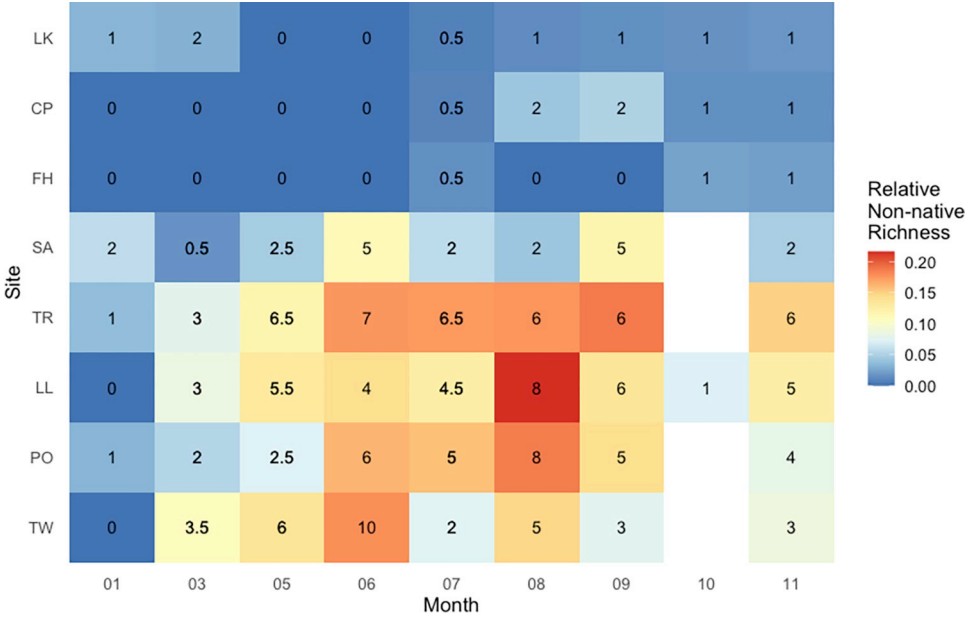

**Fig 4. Relative and absolute non-native richness across months and sampling location.** Numbers in black are the mean absolute non-native species richness at a site if it was sampled in both 2017 and 2018, or simply the absolute non-native species richness if it was only sampled in 2017.

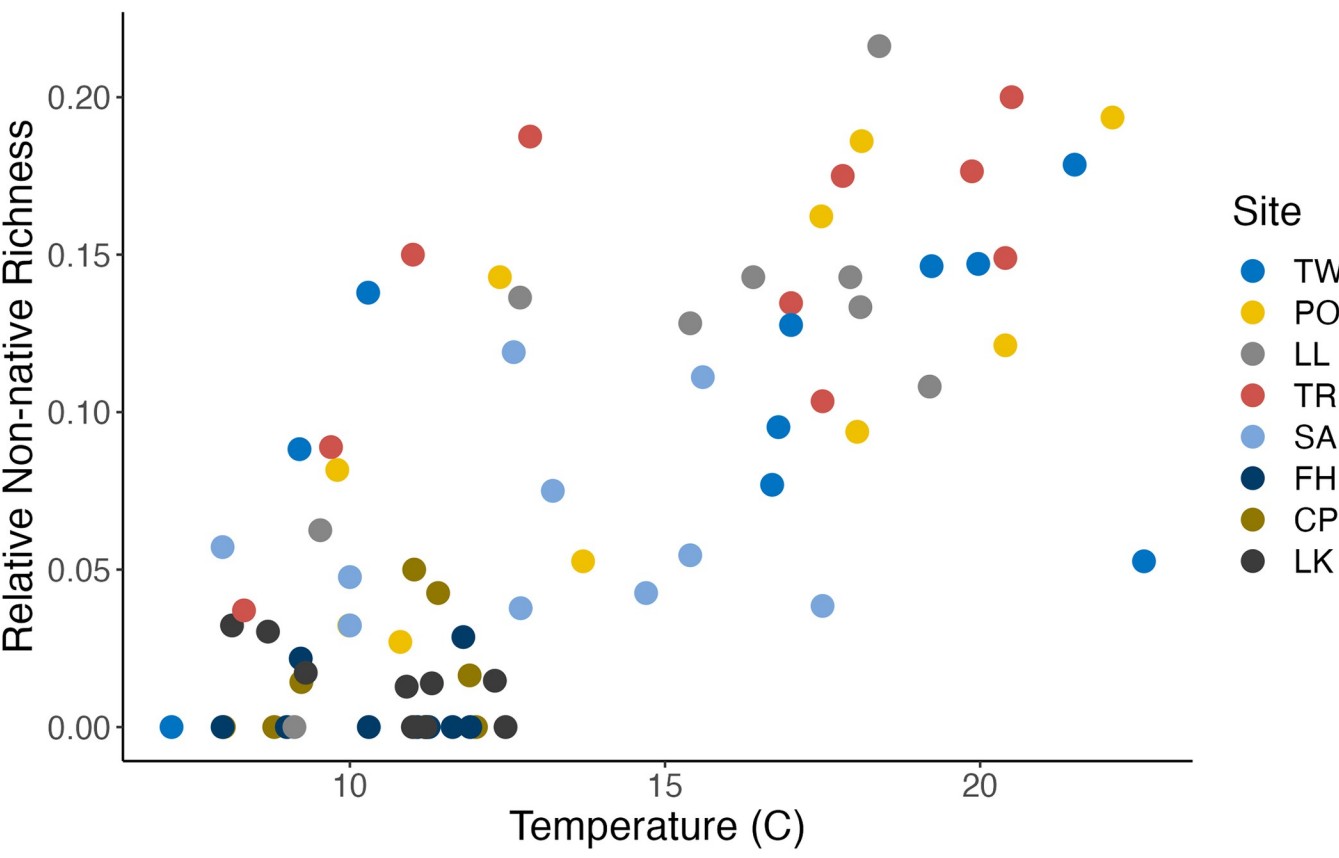

**Fig 5. Variation in relative non-native richness with temperature.** Each point represents a sampling event. Sampling sites are differentiated by color.

max = 78). Candidate Poisson regression models of non-native species richness included combinations of salinity, temperature, and native species richness as predictive variables. The best-fit model included just temperature and native species richness (**Table 3** **and** **Fig 7**). Within the selected model, higher temperatures were associated with greater non-native species richness (0.122 mean slope estimate, [0.101–0.145] 10% - 90% credibility interval), while greater native species richness was associated with lower non-native species richness (-0.011, [-0.018 –-0.005]) (**Table 4** **&** **Fig 7**). Given these parameter estimates, we would expect to observe, on average, one additional non-native species with an increase in water temperature from 15 C to 17.5 C–consistent with anticipated near-term changes in temperate regions [63]–under conditions similar to those sampled here. This would represent, at a minimum, an increase in non-native species richness of 10% relative to the Hood Canal communities–and a greater relative increase at sites with a lower baseline richness of non-native species.

## Discussion

The results of our survey demonstrate varying levels of invasion between habitat types in the Salish Sea. The more marine San Juan Island habitats show negligible levels of invasion. The mesohaline estuarine mudflats of Hood Canal show greater levels of invasion, driven primarily by the number of non-native species of algae and bivalves. The transitional habitat at Salisbury at the mouth of the Hood Canal has an intermediate invasion level. These results are consistent with known dynamics in invasion ecology—that more-retentive estuarine habitats exhibit higher levels of invasion than adjacent open coasts and that increasing temperatures make

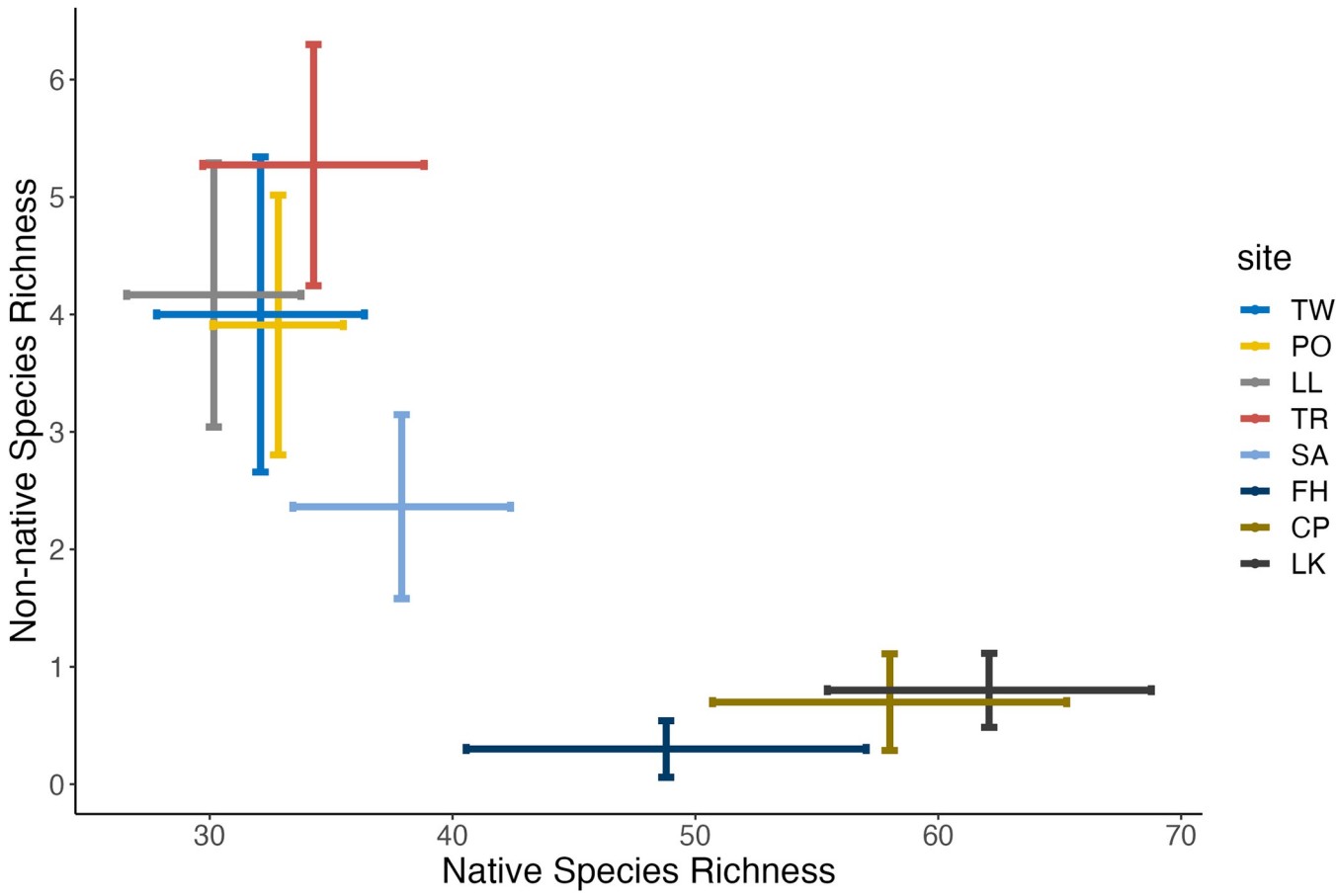

**Fig 6. The mean non-native and native species richness at the eight sampling sites.** Error bars correspond to +/- 1 standard deviation. The color gradient reflects the rough gradient from warmer, more-retentive estuarine mudflats (red) to cooler, less-retentive rocky bench (blue).

estuarine habitats more vulnerable to invasion [31, 40, 64–66]. Our results are also consistent with expectations of the biotic resistance hypothesis—that habitats with greater native species richness may resist invasion [36, 66].

## Non-native species detected

Our survey contributes new information relevant to the monitoring of non-native and invasive species in the Salish Sea, and detects non-native species previously unreported within the Salish Sea. *Gracilaria vermiculophylla* is a red alga whose native range encompasses the western North Pacific. It is a known invader of temperate estuaries around the world and has been

**Table 3. Leave-one-out model selection including parameters, difference in expected log pointwise predictive density (ELPD) and difference in Standard Error (SE) of the ELPD estimate.**

| Model | Δ ELPD | Δ SE |
|---|---|---|
| temperature, native richness | 0 | 0 |
| temperature, salinity | -0.8 | 3.2 |
| temperature | -0.9 | 2.6 |
| temperature, salinity, native richness | -1.1 | 1.6 |
| native richness | -27.0 | 11.2 |

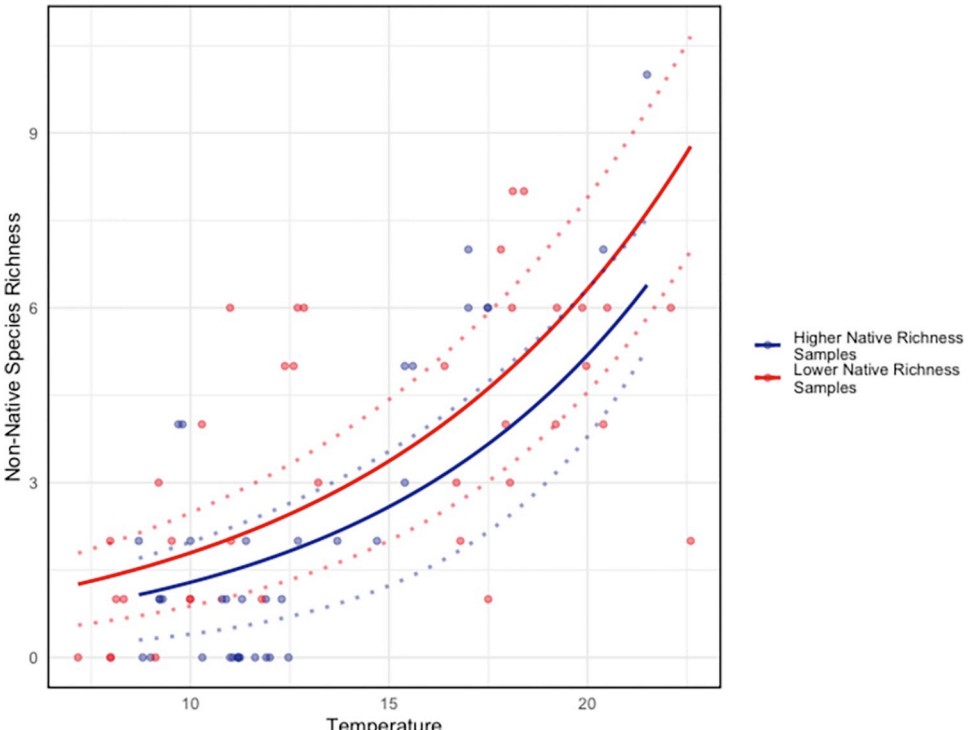

**Fig 7. Predictions from the selected model—with temperature and native species richness as parameters** (Table 4). Predictions are plotted as a function of temperature, the dominant effect. Data is subdivided into "Higher Native Richness" (n = 37) and "Lower Native Richness" (n = 39), by the 50th percentile among all sampling events.

documented in the eastern Pacific from Mexico to Alaska [67]. The ecological impacts of *Gracilaria vermiculophylla* are mixed. It has been demonstrated to out-compete native algae, but also to increase habitat complexity [68]. It has previously been identified in the Salish Sea by both visual and molecular methods. Possible vectors of introduction include hull fouling of commercial vessels, transport associated with the aquaculture oyster *Crassostrea gigas*, and transport in ballast water [67, 69]. *Gracilaria vermiculophylla* is known to tolerate mesohaline conditions, which is consistent with our findings where it was present in 34 of 86 samples, exclusively in Hood Canal.

*Caulacanthus okamurae* is a red alga whose native range encompasses the western Pacific. It is a known invader of the eastern Pacific, detected in Baja California, Mexico in 1944 and in Prince William Sound, Alaska in 1989. It has been documented on San Juan Island and the nearby Strait of Georgia [58]. We detected *C. okamurae* at all 5 Hood Canal sites, and in 23 of 64 samples. We did not detect it on San Juan Island. In California, *C. okamurae* has increased total turf cover and overall algal and invertebrate diversity in intertidal systems. It was found

**Table 4. Description of the best-fit model of non-native species richness.** Table contains details of the credibility intervals for the posterior distributions of the model's coefficients.

| parameter | mean | std dev | 10% | 50% | 90% |
|---|---|---|---|---|---|
| (Intercept) | -0.275 | 0.410 | -0.807 | -0.265 | 0.254 |
| Temperature | 0.122 | 0.017 | 0.101 | 0.122 | 0.145 |
| Native Richness | -0.011 | 0.005 | -0.018 | -0.011 | -0.005 |

to displace macroinvertebrates while favoring copepods, ostracods and other algae including *Ulva spp.*, *Chondracanthus spp.*, and *Gelidium spp.* [58, 70].

*Botrylloides violaceus* is a colonial ascidian native to the western Pacific. It is a well-documented invader of the eastern Pacific and was detected by visual survey in the southern Salish Sea in 1998 [71]. *B. violaceus* outcompetes both native and non-native fouling species including native ascidians and mussels [72]. It fouls aquaculture equipment and impacts recruitment of aquaculture species [73].

Species captured by our sampling but previously unreported in the Salish Sea include *Pyropia haitanensis*, a red alga native to the western Pacific where it is widely cultivated. It makes up the majority of all Chinese cultivation of *Pyropia spp.* [74]. The taxonomy of the family of seaweeds that includes *Pyropia spp.*, *Porphyra spp.*, and *Neoporphyra spp.* remains an area of active research. There are no prior visual records or herbarium specimens of *P. haitanensis* or its historical synonyms in the eastern Pacific or the Salish Sea. A metabarcoding survey of seaweed communities in the northern Gulf of Mexico detected *P. haitanensis* using an 23S rDNA assay, distinct from COI. Little is known about the ecological impacts of non-native *Pyropia spp.* We also detected *Callithamnion corymbosum*. There are no prior visual records or herbarium specimens of *C. corymbosum* in the Salish Sea. *C. corymbosum* is a red alga native to the temperate Atlantic. *C. corymbosum* was documented by visual survey at Willapa Bay on the outer coast of Washington State in 2008 [75]. There is no published literature on the impacts of *C. corymbosum* on native communities.

## Predictors of invasion level

**Temperature.** We observed an increase in invasion level with increases in temperature across the range of habitat types and conditions represented in our samples. Previous mesocosm and observational studies generally have found non-native fouling species to be more tolerant of high temperatures than their native counterparts. Field studies have found increased recruitment of non-native ascidians—including *B. violaceus*—to be positively correlated with higher water temperatures, and native recruitment to be negatively correlated with higher temperatures [12, 65]. In Long Island Sound, USA the invasive alga *G. vermiculophylla* was found to have higher growth rates at higher temperatures relative to the native *Gracilaria tikvahiae* [12]. Among the Poisson regression models we tested, the model including temperature and native species richness was found to be the best predictor of non-native species richness. Temperature was the dominant factor (**Fig 7**). This result provides additional evidence, at an ecosystem scale, for the positive response of non-native species, relative to native species, to higher temperature conditions.

**More-retentive estuary vs. better-flushed coast.** Globally, temperate estuaries experience higher levels of invasion than adjacent better-flushed coastal habitats [31–33]. Ours is the first survey to compare levels of invasion in a more-retentive estuary with better-flushed rocky habitat using eDNA methods. Our findings—higher levels of invasion at the more-retentive estuarine Hood Canal sites, intermediate levels at the transitional habitat at Salisbury, and negligible invasion levels at the better-flushed San Juan Island sites—are consistent with the findings of conventional surveys. Multiple mechanisms proposed by previous investigators may account for these results.

Oceanographic circulation likely plays a role. Limited circulation in estuaries may increase the likelihood that newly introduced species will be retained long enough to successfully reproduce [32]. Our results are consistent with this hypothesis: invasion levels are highest in the more-retentive environment of Hood Canal and lowest in the less-retentive environment of San Juan Island, where water circulation is stronger [76].

**Aquaculture.** Our findings may be driven by the history of aquaculture in our study region. It is probable that the Hood Canal sites experienced greater non-native propagule pressure in the form of historical transport of shellfish for aquaculture, along with the non-native algae and parasites accompanying these shellfish. Dabob Bay is an inlet in the Hood Canal situated between the Salisbury and Triton Cove sites (**Fig 1**). Between 1919 and 1935 Dabob Bay was a destination for unregulated shipments of *Crassostrea gigas* oyster spat on discarded shells originating from Japan [77, 78]. The escape and establishment of non-native species from aquaculture operations is a well-documented phenomena globally [41, 79, 80]. The high number of Manilla clam and Pacific oyster detections in our data, as well as the high proportion of species native to the temperate western Pacific, are consistent with this mechanism. Of all species detected, 15 of 21 have native ranges in Japan and the western Pacific. Among the algae, 7 of 9 have native ranges including Japan. *Mytilicola orientalis*—native to the western Pacific—is a copepod parasite of *Crassostrea gigas*.

**Anthropogenic disturbance.** Anthropogenic disturbances are known to facilitate biological invasions in marine environments. Such disruptions may make habitat less suitable for native species while also creating space and freeing-up resources for non-native species. Shoreline armoring may facilitate the spread of non-native seaweeds [81]. *Musculista senhousia* is more successful in disturbed eelgrass beds than undisturbed beds [82]. Additionally, the disturbance caused by the introduction of a single fouling species can facilitate the establishment of additional non-native species and result in an invasional cycle. Such a cycle has been documented in the Gulf of Maine facilitated by the Japanese alga *Codium fragile spp.* [83, 84]. Hood Canal has a far greater percentage of shoreline armoring than San Juan Island. Moreover, the historical aquaculture trade may have served as the catalyst for an invasional cycle in addition to creating propagule pressure. In sum, the relatively high frequency and intensity of shoreline hardening and aquaculture in the Hood Canal could contribute to the higher level of invasion recorded there.

**Biotic resistance.** Consistent with numerous previous investigations [35, 85], we found lower native species richness at soft-bottom mesohaline and polyhaline habitats and greater native species richness at more marine rocky bench habitats. Our best-fit model of non-native species richness includes both temperature and native species richness. Notably, this model outperforms the model that includes temperature alone. This finding aligns with previous investigations in which high native species diversity has been correlated to lower levels of invasion, but is eclipsed by other factors—often traits of the invaders, or abiotic filters [29, 86]. An observational study specific to fouling communities by Stachowicz & Byrnes (2006) suggested that non-native richness in these communities is negatively correlated with native richness only if space is limited and native species that facilitate non-native species settlement are rare [66].

## Detection and annotation with eDNA

As with other methods of non-native species monitoring, eDNA surveys are subject to false positives, false negatives, and biases [87]. eDNA shedding and decay rates vary across species, life-stages, and environmental conditions [88]. Primer bias may lead to inconsistent probabilities of detection across taxa [89]. For tracking ongoing biological invasions, open-source sequence data offer an advantage over visual data in that they can be amended as genetic reference databases improve. However, the degree to which eDNA assemblages reflect the underlying community depends on the quality of these databases [90]. We cannot use metabarcoding to identify species that are not in these databases.

Due to primer bias, the absence of a species at a sampling site cannot be inferred from the absence of an ASV in a water sample. For example, the invasive western Pacific alga *Sargassum*

*muticum* can be visually identified at many of our sampling sites, but *S. muticum* DNA was never identified in the water samples. However, given the diversity of life histories and morphologies represented among both native and non-native species in our study region, there is no reason to assume a systematic bias towards detection of either native or non-native species. Therefore, inferences about invasion level and non-native species assemblages are as likely to reflect underlying ecological processes as inferences drawn from visual survey data.

## Conclusion

Our results show that within the Salish Sea, more-retentive estuarine habitats with mid-range salinities exhibited higher levels of biological invasion than less-retentive, less-estuarine habitats, where invasion levels were comparatively low. Across habitats and seasons, higher water temperatures and lower native species richness were associated with higher non-native species richness. The significant positive relationship observed between temperature and invasion level suggests that habitats may be most vulnerable to invasion during seasonal windows in the spring and summer, and more importantly, suggests that habitats in the Salish Sea may become increasingly vulnerable to biological invasion as anthropogenic climate change expands seasonal windows and causes waters to warm.

The species-level data generated by this survey may help guide ongoing monitoring efforts, especially for species that were previously undetected in the Salish Sea, or are difficult to differentiate from native relatives by visual means, such as the algae *Gelidiophycus freshwateri* and its native cousins, or *Pyropia haitanensis* and native *Pyropia* species. Our data provide a rough "when and where" for future monitoring and control efforts. Using eDNA to estimate invasion level provides additional information to help guide management decisions, including those regarding the utility of prevention versus containment measures. eDNA surveys can consistently sample both non-native and native species at an ecosystem scale, offering a promising new stream of data for illuminating regional invasions, and for investigating foundational questions in invasion ecology.

## Supporting information

**S1 File.**
(DOCX)

## Author Contributions

**Conceptualization:** Joe Duprey, Ramón Gallego, Terrie Klinger, Ryan P. Kelly.

**Data curation:** Joe Duprey, Ramón Gallego, Ryan P. Kelly.

**Formal analysis:** Joe Duprey, Ryan P. Kelly.

**Investigation:** Joe Duprey, Ramón Gallego.

**Methodology:** Joe Duprey, Terrie Klinger, Ryan P. Kelly.

**Resources:** Ramón Gallego, Terrie Klinger.

**Supervision:** Terrie Klinger, Ryan P. Kelly.

**Validation:** Terrie Klinger, Ryan P. Kelly.

**Visualization:** Joe Duprey.

**Writing – original draft:** Joe Duprey.

**Writing – review & editing:** Joe Duprey, Ramón Gallego, Terrie Klinger, Ryan P. Kelly.

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
