## [Decision Letter · Decision Letter 0]

11 Jul 2023

PONE-D-23-02132Environmental DNA reveals patterns of biological invasion in an inland seaPLOS ONE

Dear Dr. Duprey,

Thank you for submitting your manuscript to PLOS ONE. After careful consideration, we feel that it has merit but does not fully meet PLOS ONE’s publication criteria as it currently stands. Therefore, we invite you to submit a revised version of the manuscript that addresses the points raised during the review process.

We look forward to receiving your revised manuscript.

Kind regards,

Hideyuki Doi

Academic Editor

PLOS ONE

Journal Requirements:

3.  We noted in your submission details that a portion of your manuscript may have been presented or published elsewhere:

"The sampling protocol and bioinformatic pipeline that generated data for this manuscript was also published in Gallego et al. (2020), and is cited appropriately throughout. This does not constitute a dual publication because the data on non-native species assemblage and invasion level presented in this manuscript has not been published elsewhere. "

Please clarify whether this publication was peer-reviewed and formally published. If this work was previously peer-reviewed and published, in the cover letter please provide the reason that this work does not constitute dual publication and should be included in the current manuscript.

5. We note that Figure 1 in your submission contain map images which may be copyrighted. All PLOS content is published under the Creative Commons Attribution License (CC BY 4.0), which means that the manuscript, images, and Supporting Information files will be freely available online, and any third party is permitted to access, download, copy, distribute, and use these materials in any way, even commercially, with proper attribution. For these reasons, we cannot publish previously copyrighted maps or satellite images created using proprietary data, such as Google software (Google Maps, Street View, and Earth). For more information, see our copyright guidelines: http://journals.plos.org/plosone/s/licenses-and-copyright.

(1) You may seek permission from the original copyright holder of Figure 1 to publish the content specifically under the CC BY 4.0 license.  

**Additional Editor Comments:**

I got the recommendations and comments from an expert reviewer in the field. The reviewer agreed that the manuscript is technically sound and the data support the conclusions. However, the lack of an explanation in the Methods sections and figure presentations were suggested by the reviewer, and I share the comments. Therefore, I can invite you to submit a revised version of the manuscript that addresses the points raised by the reviewer.

Reviewers' comments:

Reviewer's Responses to Questions

**Comments to the Author**

1. Is the manuscript technically sound, and do the data support the conclusions?

Reviewer #1: Yes

2. Has the statistical analysis been performed appropriately and rigorously? 

Reviewer #1: Yes

3. Have the authors made all data underlying the findings in their manuscript fully available?

Reviewer #1: Yes

4. Is the manuscript presented in an intelligible fashion and written in standard English?

Reviewer #1: Yes

5. Review Comments to the Author

Reviewer #1: The study entitled “Environmental DNA reveals patterns of biological invasion in an inland sea” study which habitat types are most prone to biological invasion and what are the vector that drive those invasions. I am not an expert on environmental DNA but I think that the paper is written in a clear manner and it is easy to follow the flow of the research article. Beside that, I’ve made some suggestion that, in my opinion, can strength the paper.

I think that in some part this paper can be improve with some work on the images. In the detailed comments I have made different advice.

To strengthen the paper, I suggest:

1. To Improve the method part as suggested in the detailed comments.

2. To improve the readability of the figure created.

DETAILED COMMENTS:

• L123-125 repetition of concept, I suggest rewrite the sentence to make it clearer.

• L186-188 Is there any reference on this approach used? Any previous work?

• L202 you should cite both R and the R package used.

• L211 cite the vegan package.

• Figure 1 is missing the scale and the northern arrow. I also suggest using different symbol instead of colors to point the study sites.

• Figure 5 is it possible to have symbols instead of colors? The graph with color is pretty hard to read.

• Figure 6 if you use color to define the study site you can’t change the color you assigned to another figure. For example, at TW is assigned red but in the fig 5 is the blue one. That make the readability of the graph lower.

6. PLOS authors have the option to publish the peer review history of their article (what does this mean?). If published, this will include your full peer review and any attached files.

Reviewer #1: No

---

## [Author Response · Author response to Decision Letter 0]

14 Nov 2023

also see attached response_to_reviewers.docx:

We appreciate the comments made by the reviewer and have addressed them as described below. 

L123-125 repetition of concept, I suggest rewrite the sentence to make it clearer.

I’ve eliminated the repeated concept from the previous paragraph L114-115. I’ve also clarified the comparison of abiotic habitat factors between sites L124-L126.

L186-188 Is there any reference on this approach used? Any previous work?

To our knowledge no previously published work has taken this exact approach to classifying species as native, non-native or cryptogenic. Previously published invasive species research often reports similar approaches, or does not explicitly report their classification process. We offer examples from two of the non-native species metabarcoding studies cited in our manuscript: 

Lacoursiere-Roussel et al (2018) state that they identified potential “NIS invaders” based on previously published risk assessments, and monitoring studies of their study region – an approach similar to our own. They also highlight that “focal taxa [potential NIS invaders] vary among surveys”, an observation that highlights the challenges of this classification process in general.

Holmen et al (2019) do not explicitly state how they classify species as non-native to their study region. “In total 18 NIS to the study region and 24 species documented as NIS in other regions were detected across the four sites (see Supplementary Table 2 for full list). Out of the detected NIS, eight were present in the list of 21 NIS previously detected in rapid assessment (RA) surveys at the sampling sites”.

We note that in the context of the broader non-native species literature our approach to classifying species is conservative – relying on multiple databases sourced from peer reviewed literature, and requiring multiple peer-reviewed sources to have already classified a species as non-native. 

L202 you should cite both R and the R package used.

Citations added for both.

L211 cite the vegan package.

Citation added. 

Figure 1 is missing the scale and the northern arrow. I also suggest using different symbol instead of colors to point the study sites.

Fig1: I have added a scale bar and a north arrow. I have made the study sites different symbols in addition to different colors to represent that different substrate types. 

Figure 5 is it possible to have symbols instead of colors? The graph with color is pretty hard to read.

Fig5: I think symbols for each site distract from the main take-away of this scatter plot which is the relationship between temperature and relative non-native richness. I have increased the size of each data-point to make the color more legible. 

Figure 6 if you use color to define the study site you can’t change the color you assigned to another figure. For example, at TW is assigned red but in the fig 5 is the blue one. That make the readability of the graph lower.

Fig 6: I have changed the colors on Figure 6 so that each site has the same color on both Figure 5 and Figure 6.

---

## [Editor Report · Decision Letter 1]

29 Nov 2023

Environmental DNA reveals patterns of biological invasion in an inland sea

PONE-D-23-02132R1

Dear Dr. Duprey,

We’re pleased to inform you that your manuscript has been judged scientifically suitable for publication and will be formally accepted for publication once it meets all outstanding technical requirements.

Kind regards,

Hideyuki Doi

Academic Editor

PLOS ONE

Additional Editor Comments (optional):

I carefully checked the revised manuscript as well as the response letter. I agree with the revisions and now can recommend publishing the paper.
---

## [Editor Report · Acceptance letter]

6 Dec 2023

PONE-D-23-02132R1 

Environmental DNA reveals patterns of biological invasion in an inland sea 

Dear Dr. Duprey:

I'm pleased to inform you that your manuscript has been deemed suitable for publication in PLOS ONE. Congratulations! Your manuscript is now with our production department. 

Kind regards, 

on behalf of

Dr. Hideyuki Doi 

Academic Editor

PLOS ONE